# Trends in Viral Vector-Based Vaccines for Tuberculosis: A Patent Review (2010–2023)

**DOI:** 10.3390/vaccines12080876

**Published:** 2024-08-02

**Authors:** Lana C. Santos, Antônio Márcio Santana Fernandes, Izabel Almeida Alves, Mairim Russo Serafini, Leandra da Silva e Silva, Humberto Fonseca de Freitas, Luciana C. C. Leite, Carina C. Santos

**Affiliations:** 1Serviço de Imunologia das Doenças Infecciosas, Faculdade de Farmácia, Universidade Federal da Bahia, Salvador 40170-115, BA, Brazil; lana_cs@outlook.com (L.C.S.); marciofernandes14@gmail.com (A.M.S.F.); leandra.silva@ufba.br (L.d.S.e.S.); 2Departamento do Medicamento, Faculdade de Farmácia, Universidade Federal da Bahia, Salvador 40170-115, BA, Brazil; izabel.alves@ufba.br; 3Programa de Pós-Graduação em Ciências Farmacêuticas, Universidade do Estado da Bahia, Salvador 41150-000, BA, Brazil; 4Departamento de Farmácia, Universidade Federal do Sergipe, São Cristóvão 49100-000, SE, Brazil; maiserafini@hotmail.com; 5Departamento de Saúde, Universidade Estadual de Feira de Santana, Feira de Santana 44036-900, BA, Brazil; hffreitas@uefs.br; 6Laboratório de Desenvolvimento de Vacinas, Instituto Butantan, São Paulo 05503-900, SP, Brazil; luciana.leite@butantan.gov.br; 7Departamento de Análises Clínicas e Toxicológicas, Faculdade de Farmácia, Universidade Federal da Bahia, Salvador 40170-115, BA, Brazil

**Keywords:** tuberculosis, vaccine, patents, subunit, viral vector

## Abstract

Tuberculosis (TB) is an ancient global public health problem. Several strategies have been applied to develop new and more effective vaccines against TB, from attenuated or inactivated mycobacteria to recombinant subunit or genetic vaccines, including viral vectors. This review aimed to evaluate patents filed between 2010 and 2023 for TB vaccine candidates. It focuses on viral vector-based strategies. A search was carried out in Espacenet, using the descriptors “mycobacterium and tuberculosis” and the classification A61K39. Of the 411 patents preliminarily identified, the majority were related to subunit vaccines, with 10 patents based on viral vector platforms selected in this study. Most of the identified patents belong to the United States or China, with a concentration of patent filings between 2013 and 2023. Adenoviruses were the most explored viral vectors, and the most common immunodominant *Mycobacterium tuberculosis* (Mtb) antigens were present in all the selected patents. The majority of patents were tested in mouse models by intranasal or subcutaneous route of immunization. In the coming years, an increased use of this platform for prophylactic and/or therapeutic approaches for TB and other diseases is expected. Along with this, expanding knowledge about the safety of this technology is essential to advance its use.

## 1. Introduction

*Mycobacterium tuberculosis* (Mtb) is a globally distributed agent. It is transmitted through the inhalation of aerosolized droplets containing bacteria, causing tuberculosis (TB), a disease extensively studied worldwide [1]. Although primarily affecting the lungs, with the pulmonary form being the most common clinical manifestation of the disease, TB is a multisystemic disease that can affect organs such as the brain, intestines, kidneys, spine, and lymph nodes [2].

According to the World Health Organization (WHO), about 25% of the world’s population shows immunological evidence of prior Mtb infection, resulting in over 10 million new cases and the death of 1.3 million individuals annually [3]. From 2016 to 2020, Brazil was listed by the WHO among countries with a high burden of TB, and TB/Human Immunodeficiency Virus (HIV) coinfection [3], together with malaria and HIV/Acquired immunodeficiency syndrome (AIDS), represents one of the deadliest infections worldwide, bringing significant socioeconomic impacts to humanity [4].

Treating TB continues to be a challenge. Treated patients may face pulmonary sequelae and respiratory complications even after recovery from the disease [5]. Due to limitations in available treatments, administration of several medications over several months is recommended. A careful analysis needs to be made for this recommendation while considering the assessment of the balance between treatment efficacy and the challenges associated with duration, complexity, and toxicity [6].

The spread of drug-resistant Mtb strains poses a challenge to public health. This may require the combination of different medications, which generate undesirable effects for the patient, such as potential nephrotoxicity [6,7]. Additionally, the costs associated with treating patients with drug-resistant TB are high. In 2022, the estimated proportion of people with TB who had multidrug-resistant TB/rifampicin-resistant TB (MDR/RR-TB) was about 3.3% among new cases and 17% among those previously treated [3].

There is broad consensus, supported by epidemiological models, which indicates that it is essential to develop new highly effective TB vaccines that can interrupt the transmission cycle to achieve the goals, by 2035, set by the WHO [3,8]. These goals aim to reduce TB-related deaths by up to 95% and the disease incidence rate by 90%, as compared to the 2015 levels [3]. The attenuated live vaccine *Mycobacterium bovis* bacille Calmette–Guérin (BCG) has been applied against TB worldwide for over a century, with over 100 million children vaccinated annually [9,10]. Clinical trials have shown that infant BCG has moderate efficacy against severe extrapulmonary forms of TB [11]. In pulmonary form, efficacy among adolescents and adults ranges from 0% to 80%. Although it is generally considered safe, the use of this vaccine in immunocompromised individuals is controversial [12,13]. This reinforces the need for additional strategies in employing novel vaccine technologies that generate longer-lasting and safer immune responses while safeguarding against Mtb infection [14,15,16].

Recent decades have seen significant advances in the development of TB vaccines. Many of these vaccines are in various stages of preclinical or clinical trials and employ technological strategies such as the following: (i) Inactivated or fragmented mycobacteria; (ii) Recombinant BCG (rBCG) vaccines; (iii) Protein subunit vaccines as combined with different adjuvants; (iv) Viral vector vaccines [17,18,19,20,21]. The COVID-19 pandemic accelerated the industrial interest, development, and application of several technological platforms that have been little explored or moving slowly, such as mRNA and viral vector-based vaccines, and provided important insights for tuberculosis vaccine research and development [22,23].

Viral vector vaccines are platforms designed to overexpress antigens and elicit immune responses without the need for adjuvants [24,25]. The signaling mechanisms of viral vectors involve the activation of a pro-inflammatory response, including the production of cytokines and chemokines, promoting both humoral and cellular immune responses without causing excessive cytokine production, which can harm the host. The most used viral vectors for TB are derived from vaccinia virus, adenovirus, and influenza virus, which are easy to design, safe, and immunogenic in humans [24,25]. They do not necessarily require storage at very low temperatures, making them an ideal platform for equitable global distribution or storage [26]. Given the renovated interest in this type of platform in recent years, the aim of this review was to evaluate the published patents for the development of vaccine candidates for TB (between 2010 and 2023) while focusing on strategies based on viral vectors.

## 2. Materials and Methods

This patent review was conducted in Espacenet, a database of the European Patent Office (EPO), using the descriptors “mycobacterium and tuberculosis” and the classification A61K39, which is related to medical preparations containing antigens or antibodies. The preliminary selection was based on inclusion criteria: patents published from January 2010 to December 2023, in any language, containing the descriptors in the title or abstract, and considering the classification used in the search as mentioned previously. Around 411 patents were initially identified, of which 10 duplicates were removed (Figure 1).

Careful analysis of the titles and abstracts reduced the selection by another 216 for the following reasons: no indication for use as vaccine or only proposed as diagnostic methods or treatments, indication as vaccines for other pathogens (not Mtb), suggested as attenuated or inactivated vaccines, proposed as vaccines based on nucleic acids (DNA, RNA) or based on rBCG. Four patents were excluded due to unavailability of translation. At this stage, only subunit vaccines were included, leaving 181 patents. By thoroughly reading the patents, only those related to viral vector vaccines were selected, comprising a total of 10 patents (Figure 1).

Using the same criteria, a search was also conducted on the Scopus platform to compare the number of articles with the number of patents identified in this review. This investigation was carried out in April 2024. Regarding the patents, we adhered to the Preferred Reporting Items for Systematic Reviews and Meta-Analyses (PRISMA) guidelines for the search and screening. Additionally, the contents of the patents were categorized by year and country.

## 3. Results and Discussion

### 3.1. Main Patented Strategies for TB Vaccine between 2010 and 2023: Few Patents Based on Nucleic Acid, Viral Vector, or Inactivation

Through patent screening, we observed that the majority of patents identified were related to subunit vaccines (proteins or fragments) (n = 136), rBCG (n = 32), followed by live attenuated (n = 16) and acid nucleic vaccines (n = 16). A small number of patents were associated with vaccine platforms based on viral vectors (n = 10) and inactivation (n = 6) (Figure 2A). Among subunit vaccines, we observe at least two patents published in recent years based on the virus-like particle strategy. Interestingly, a patent identified in our search, the US2023365631 (A1), provided a wide application of Mtb antigens in different platforms, such as rBCG, viral vector, and acid nucleic.

In a recently published review, the authors provide data on 19 vaccine candidates for TB in clinical development; three were identified with therapeutic and 14 with prophylactic approaches [19]. According to this study, the highest number of vaccines that progressed to clinical trials are those utilizing the subunit platform (6), followed by viral vector (5), inactivated vaccines (4), and, in smaller numbers, live attenuated vaccines (2), rBCG and nucleic acids (1 each). Another recent review highlights advances in clinical trials of subunit and viral vector vaccines for TB [27]. This is in line with our findings in this review, as the majority of patents published in the period 2010–2023 were based on the subunit strategies, although it is notable that fewer were related to viral vector approaches.

Inactivated and live attenuated vaccines are identified in the first generation of vaccines and have been widely applied to different diseases, such as TB, although their production and safety can pose challenges to their use. The development of the recombinant technology changed this scenario, increasing interest in the design of subunit vaccines, identified in the second-generation vaccines. Extensive advances in molecular genetics over the past forty years have sparked interest in viral vector strategies. The development of adjuvants has also facilitated the use of these technologies [18,19,25,26,27,28]. In this context, our focus in this review was to address those based on viral vectors, identifying the main Mtb antigens that have been used and experimental models used for their evaluation.

### 3.2. Patents Based on Viral Vector for TB between 2010 and 2023

Of the ten selected patents, four are registered in the United States of America (US) and two in the China Patent Office, as shown in Figure 2B. These countries are recognized for having a robust patent registration culture, likely due to their strong economy and considerable investments in technology and innovation, placing them in a prominent position in terms of intellectual property in technological research [29]. Russia and South Korea presented one patent each. Additionally, two of the patents belong to the World Intellectual Property Organization (WIPO), a global forum in intellectual property (IP). The WIPO registration allows for multiple jurisdictions through the Madrid System, making it possible to register in several countries at the same time in a single application.

The number of patents based on viral vectors for TB registered between the years 2013 and 2023 is notable. However, there were no patents filed based on this strategy in the years 2015, 2017, 2019, and 2020 (Figure 2C). It is believed that, in the coming years, the proposal to use viral vectors in the context of TB will increase significantly. This is partly due to the SARS-CoV-2 pandemic, which resulted in a significant increase in the use of this platform, with millions of doses administered worldwide. Although not yet completely evaluated, this scenario provided data on the safety, immunogenicity, and efficacy of this technology [22].

Other aspects that may support this expectation are the use and expansion of knowledge on different viral vectors (little or not yet explored for TB) [26] and interest in diversifying the TB vaccine pipeline (preclinical and clinical), as highlighted by the TB Vaccine Roadmap Stakeholder Group [23]. For example, exploring new Mtb antigens or scaling up different platforms that reach advanced clinical phases, as most are live attenuated, killed, and adjuvanted protein vaccines [19,23,27]. Some vaccine candidates for TB based on viral vectors have reached the clinical trials [19,27]; one of them, the modified vaccinia Ankara/Ag85A vaccine, in a phase 2b trial, had an absence of efficacy against TB in infants [30].

To compare these results with the number of articles published in the same period, a search on the Scopus platform using the descriptors “mycobacterium,” “vaccine”, and “viral vector”, identified 22 published articles. There was a significant increase in the number of publications in this field in the years 2014 and 2019, with 03 articles each year, and countries such as the United Kingdom, US, and Canada leading this list of publications. Below, we will discuss the main viral vectors, Mtb antigens, and experimental models used to support the patents selected in this study.

#### 3.2.1. Viral Vectors

Viral vectors are promising vaccine platforms that are based on recombinant viruses to deliver the genetic sequences encoding the selected antigen(s) to the host. They have the ability to express heterologous antigens and induce antigen-specific cellular immune responses, as well as robust antibody titers [24]. Precise vector design is essential for mitigating inflammatory responses, leading to successful therapies. On the other hand, the use of viral vectors for vaccines is on the rise, driven by the characteristics of the manufacturing process and the ability for rapid distribution [26]. In this review, six out of ten selected patents proposed as vaccines for TB were based on adenoviruses. The other four platforms used were influenza virus, cytomegalovirus, Sendai virus, and arenaviruses (Table 1).

Adenoviruses are non-enveloped viruses with linear DNA genomes, widespread and endemic in nature, with over 100 serotypes [41]. In humans, adenovirus can cause symptomatic infections, such as respiratory and gastrointestinal infections. Infected individuals usually have mild symptoms of cold, conjunctivitis, and pharyngitis. Another portion of infected individuals naturally recovers [42]. In the 1990s, adenoviruses began to be explored as vaccine vectors, and since 2003, they have been evaluated in clinical trials [19,43]. In recent years, several modifications of adenoviral vectors have been explored to make them more clinically relevant. Including heterologous antigens in the capsid has proven to be a successful strategy in preclinical studies. Epitopes can be incorporated into viral structural proteins to expose them on the virus surface, enabling the induction of robust humoral response [44].

Antigens from different pathogens, such as HIV-1 [45], influenza A [46], *Plasmodium falciparum* [47], and Mtb [30,31,32,33,34,35,36,37,38,39,40,43], have already been investigated using this platform. Human type 5 adenovirus (AdHu5), the most widely used adenovirus vector until recently, has the ability to induce robust CD8+ T cell and antibody responses and is effective in generating high viral titers during the manufacturing process [25]. The human adenovirus type 35 (Ad35) vector has been extensively evaluated in HIV vaccine trials, demonstrating safety and immunogenicity [48]. In this study, we identified a patent based on chimpanzee adenovirus (AdC) vector, such as AdC68, the WO2013123579 (A1); while the CN108018298 (A), KR20200076335 (A), WO2022192163 (A1) and US2023365631 (A1) patents consist of an AdHu5 vector (Table 1). The WO2022192163 (A1) patent also includes another adenovirus, the bovine adenovirus type 3 (BAd) vector. We were unable to identify which adenovirus was used in the CN112899295 (A) patent (Table 1).

Cytomegalovirus (CMV) is a member of the herpesvirus family. They generally cause asymptomatic infections in healthy individuals. CMV can elicit robust T cell responses [49], targeting a broad range of antigens [50], and also induces substantial antibody responses post-infection [51]. Genetic engineering techniques have allowed the CMV genome to be cloned into a plasmid, generating an artificial bacterial chromosome that can be modified to express different exogenous immunogens [52].

In our search, we identified the US2021403951 (A1) patent that uses rhesus CMV (RhCMV) and human CMV (HCMV). CMV-based vectors are a versatile platform able to express multiple genes, such as those related to malaria, tuberculosis, cancer, HIV, COVID-19, and other relevant applications [52,53,54,55,56,57]. Another important aspect is that CMV does not integrate into the host genome and its vectors; after modifications that inhibit viral replication, it can be used in immunocompromised individuals [58]. In this sense, studies on virus infection and immunity have been stimulated [59].

Influenza viruses cause acute respiratory infections, occasionally triggering epidemics. While most respiratory cases are mild and related to the upper airways, some individuals face a higher risk of severe complications, particularly older people, that may result in high morbidity and mortality [60]. With the development of reverse genetics techniques applicable to negative-sense RNA viruses, influenza viruses have become an attractive option for use as viral vectors in immunization strategies against a variety of pathogens [61,62].

Influenza A and B are RNA-based viruses. Among these segments, the smallest fragment (NS) is particularly suitable for genetic manipulation, as it encodes two proteins, the NS1 and Nep [60]. NS1, a non-structural protein synthesized in large quantities during infection, demonstrated the ability to tolerate relatively long amino acid insertions. Furthermore, NS1 accumulates in the nucleus and subsequently in the cytoplasm of infected cells, resulting in a robust immune response against the inserted antigen [63,64].

The NS1 protein is localized intracellularly, and some authors argue that influenza virus NS vectors may be less efficient in inducing immunogenicity, leading to less efficient antigen presentation through the MHC class II complex [64]. However, studies demonstrate that infection of mice with influenza NS vectors can trigger CD8 T cell responses, especially when two vectors belonging to different virus subtypes are used in booster immunizations [65]. As shown in Table 1, a vaccine patent employing the influenza A (H1N1) vector platform (RU2678175 (C1)) against TB has been registered. This patent is related to the improvement of the production of TB/FLU-04L (composition already described in another patent), which is currently in clinical trials. This vaccine was designed to express Mtb antigens, the 85A antigen (Ag85A), and Early Secreted Antigenic Target 6 kDa (ESAT-6) antigens.

Sendai virus (SeV) is a type 1 parainfluenza virus, enveloped with a non-segmented negative-sense RNA genome, a member of the *Respirovirus* genus of the *Paramyxovirinae* subfamily [66,67]. Also known as the Hemagglutinating Virus of Japan or Sendai virus pneumonia (Sendai type), it was discovered in Sendai, Japan, in the 1950s [68]. Not considered a pathogen for humans, it demonstrates low pathogenicity, robust capability for expression of foreign genes, and is able to infect a wide range of hosts, in addition to serving as a vector for inducing mucosal immunity in the lungs [67,69]. Reverse genetics technology has enabled the construction of gene transfer vectors from SeV-type RNA viruses. Through this approach, developed vectors have shown significant efficacy in gene transfer and expression of foreign proteins in in vitro studies [67].

This review highlights the registration of the US10828359 (B2)/US2018085449 (A1) patent (Table 1), which describes a vaccine (SeV85AB) against TB based on replication-deficient SeV, capable of expressing the immunodominant antigens Ag85A and Ag85B from Mtb. The vaccine in question induced high levels of immune protection mediated by memory T cells resident in the lung tissue of mice in models of acute and latent infection [35].

Arenaviruses are enveloped viruses with a bi-segmented negative-sense RNA genome encoding four genes [70]. They cause chronic infections in rodents, and infections in humans are common and, in some cases, severe [71]. The use of lymphocytic choriomeningitis virus (LCMV), a type of arenavirus, as a vaccine vector was initially documented in 2009, made possible by advances in reverse genetic engineering systems, enabling the creation of recombinant LCMV (rLCMV) [67,71,72]. It is a replication-deficient viral platform, highly immunogenic, inducing broad and long-lasting T cell responses in mice and non-human primates (NHP) [73,74,75].

We highlight the registration of US2016024476 (A1)/US9809801 (B2) patent, in 2014, which describes preclinical data of a candidate TB vaccine. In mice, the rLCMV vaccine, expressing the TB10.4 and Ag85B Mtb antigens, increased the frequencies of CD8 and CD4 T cell responses and significantly reduced the lung burden of Mtb after aerosol challenge. Combining rLCMV with BCG increased immune responses to Mtb antigens encoded by rLCMV, but protection was not significantly different when compared to vaccination with rLCMV or BCG alone [34].

#### 3.2.2. Mtb Antigens

Vaccine candidates using a viral vector-based delivery platform against TB have been underexplored compared to other strategies, as shown in this study (Figure 2A). Most patents are focused on active or well-known immunodominant Mtb antigens, such as Ag85A, Ag85B, and ESAT-6 (Table 1). It is noteworthy that some of the patents published more recently (2021 and 2023), such as the US2021403951 (A1), CN112899295 (A), and US2023365631 (A1), used antigens from the active phase of TB along with antigens associated with the dormancy phase (DosR). Moreover, the US2021403951 (A1) patent included antigens from the resuscitation phase (RpfA, RpfC, RpfD). These antigens were selected due to their significant potential in inducing cytokines and stimulating T cell response and/or to play an important role in different phases of TB infection [76,77,78,79]. However, there is a need for vaccine-candidate formulations that explore other immunodominant antigens; so far, only 7% of the Mtb proteome has been explored to discover antigens able to activate robust T cell responses [79].

The Ag85 complex, which plays a crucial role in Mtb pathogenicity, is the main secretory antigen and is composed of three proteins: Ag85A (31 kDa), Ag85B (30 kDa), and Ag85C (31.5 kDa) [76]. It is involved in cell wall mycoloylation. It has a mycolyltransferase activity necessary for the biogenesis of trehalose dimycolate (cord factor), which is important for cell wall integrity [80,81]. Vaccine candidates utilizing the Ag85 complex have been shown to be immunogenic, safe, and effective, inducing robust immune responses from specific T cells in the lung and secretion of secretory immunoglobulin A (SIgA) in the lung mucosa [76].

ESAT-6 (6 kDa), much like the Ag85 complex, is an antigen secreted during the logarithmic growth of Mtb [82]. The ESAT-6 gene (esxA) is part of the esx-1 locus, a group of genes that encode the type VII secretion system, allowing secretion of the ESAT-6 virulence factor from the pathogen [83]. Considered one of the ideal candidates for subunit vaccine, recombinant protein, and viral vector vaccine due to its high immunogenicity, ease of expression, and heterologous amplification [84].

Patients and animals infected with Mtb complex strongly respond to ESAT-6 antigens. Due to its high immunogenicity, this antigen is also widely studied and included in the development of diagnostic tools for TB [85]. ESAT-6-based vaccines are extensively explored [38,86]. In this study, we highlight patent RU2678175 (C1) (Table 1), which explores a formulation using an influenza A virus vector encoding ESAT-6 and Ag85A antigens. It was demonstrated that when mice or cynomolgus monkeys were immunized with this formulation, it induced a specific Th1 immune response to Mtb. A single immunization in mice also showed high protection compared to BCG and significantly increased the protective effect of BCG when administered in a prime booster regimen [38].

In order to develop an improved TB vaccine, some studies have begun to evaluate the potential of Mtb proteins related to the resuscitation-promoting factor (Rpf), namely RpfA (Rv0867c), rpfB (Rv1009), RpfC (Rv1884c), RpfD (Rv2389c), and RpfE (Rv2450c) [77,87]. Some of these are described in the formulation of US2021403951 (A1) (Table 1). It has been shown that Rpf proteins are potent T cell antigens and induce protective immunity. They are now considered potential candidates for the development of subunit vaccines and viral vectors and as possible sources for the diagnosis of asymptomatic latent TB infection [88,89].

#### 3.2.3. Experimental Models

Data available from the patent’s description analyzed in this review showed that all the patents were evaluated in the preclinical phase, using in vitro and in vivo experimental models. Most of them used mouse models, such as BALB/c and C57BL/6; one of them, US2021403951 (A1), used an NHP model, as shown in Table 1. The strategies employed in preclinical testing of TB drugs and vaccines in animal models differ among laboratories. There is limited understanding of how variations in aerosol or intravenous infection methods, vaccine administration route, dosage, and mouse strains used in assays may produce discrepant results [90].

Mouse models are frequently used to evaluate immune response and protection against TB. The H-2 k haplotype is associated with strains resistant to active TB, while animals with other haplotypes develop a kind of chronic infection, with varying degrees of resistance among them [91]. The C3HeB/FeJ, DBA/2, 129/Sv, BALB/c, and C57BL/6 strains present the H-2k haplotype [92] and the susceptibility allele of the antimicrobial resistance gene. BALB/c and C57BL/6 strains are commonly used in TB-related studies [93]. Our results showed that at least 6 of the patents were evaluated using C57BL/6, 02 using BALB/c, and 01 using CB6F1 mice (Table 1).

The BALB/c strain is one of the animal groups that best replicates the different phases of TB; hence, it is considered the “gold standard” strain used in infection assays [94]. Studies suggest that the C57BL/6 mouse strain may behave similarly to BALB/c in infectious disease models such as TB [95]. Currently, C57BL/6 strain is the most widely used for TB studies [96]. Data from both female and male C57BL/6 mice showed that after aerosol Mtb infection, males had increased morbidity and mortality when compared to females [97].

It is interesting that the guinea pig model is considered an important confirmatory model for protection against TB since the physiology of the disease is closer to that of humans; however, none of the recent patents used this model to demonstrate the protective efficacy of the vaccine candidates [94]. This could be due to the higher costs of the essay and the lower availability of facilities to perform the Mtb challenge in the guinea pig model.

NHP, due to their similarities with humans, are attractive models for the investigation of various infectious diseases. One example is that NHP can generate clinical correlates of infection close to humans, such as cellular profiles and acute phase proteins of infection, and infection-related parameters can be assessed using imaging tests, auxiliary TB diagnostic tests, such as tuberculin skin tests, and gamma interferon release assays [98]. NHP has been used in preclinical and clinical studies of new therapeutics and vaccines [99,100]. One of the ten patents identified in our study used this model (Table 1). Although the NHP model has advantages, it requires resources, such as the need for specialized facilities and high costs, which often make its use unfeasible [101].

We observed that many of the patents were evaluated using the intranasal route of immunization (Table 1). This alternative route of immunization has been widely explored for viral vector vaccines to induce immune responses at mucosal sites against respiratory or gastrointestinal infections [102]. Considering an infectious respiratory disease like TB, it could be an interesting route of immunization. Viral vector approaches have the potential for immunization in the respiratory tract. It is known that intranasal vaccination of mice with adenoviral vectors can induce a greater immune response [103,104].

Besides that, the approach of testing the vaccine candidate alone or as a booster to the BCG vaccine has been frequently used in the selected patents. This approach has been investigated not only in the context of human TB but also in bovine TB (bTB), which is a cause for concern on the global stage as well. Although no vaccine currently offers greater protection than BCG against bTB, when used in combination with BCG, several offer improved protection, for example, mycobacterial antigens vectored by recombinant human adenovirus [105,106].

#### 3.2.4. Current Stage of Patent’s Development: Information on Preclinical and Clinical Trials

We seek information about the current stage of development of the patents identified in this review, for example, whether they are progressing to clinical trials. This information was not necessarily available in the patent description; thus, it is based on what we could find in the literature or in databases of TB vaccines and clinical trials.

As shown in Table 1, the WO2013123579 (A1) patent, based on chimpanzee adenoviruses (AdC), is in the preclinical stage [31,32,33]. It was developed by the same group that developed the AdHu5Ag85, which is currently in phase I clinical trials (NCT02337270). Although AdHu5Ag85 has demonstrated immunogenicity and protection in preclinical and clinical trials [107,108], an important limitation to application in humans is pre-existing immunity to AdHu5 due to respiratory exposure to the AdHu5 virus. This may compromise the potency of the vaccine and its safety when administered to populations at high risk for HIV [31,108]. Alternatively, a vaccine based on the AdC vector has been developed, which rarely elicits a pre-existing antibody response in humans [31,109,110] and uses the same cellular entry receptors as AdHu5 [31].

Among the other adenovirus-based patents, the WO2022192163 (A1) patent, which comprises vaccine candidates expressing the autophagy-inducing peptide C5 and mycobacterial Ag85B-p25 epitope using human (HAdv85C5) and bovine (BAdv85C5) adenoviruses vectors, are in preclinical phase [40]. It is noteworthy that pre-existing humoral immune responses against adenovirus do not interfere with the immunogenicity of BAdv vector-based vaccines [40,111] and that BAdv85C5 is a promising mucosal vaccine for tuberculosis [40]. This is one of the most recently registered patents identified in this review (Table 1). No references were found in the literature or in clinical trial databases related to patents CN108018298 (A) and KR102135334 (B1)/KR20200076335 (A), also based on AdHu5, nor to CN112899295 (A), identified in this review. Based on the information contained in the patent descriptions, they are in the preclinical stage (Table 1).

The US2023365631 (A1) patent, the newest registered patent identified in this review (2023), describes Mtb antigens expression in many viral vectors, such as AdHu5, chimpanzee adenovirus, poxvirus, rhesus (RhCMV) and human cytomegalovirus (HCMV). Based on the information contained in the patent, the candidate vaccines are in the preclinical phase, and no published references were found. Another patent based on RhCMV or HCMV vectors identified in this review, the US2021403951 (A1), is in the preclinical phase [39] (Table 1).

The US2016024476 (A1)/US9809801 (B2), US10828359 (B2)/US2018085449 (A1), and RU2678175 (C1) patents are based on RNA viral vectors, arenaviruses, Sendai and influenza viruses, respectively. RNA viral vectors have some advantages: There is the prolonged expression of proteins in high levels, they may meet stringent safety concerns, and there is no concern about the integration of foreign sequences into chromosomal DNA [26,112,113]. So far as we know, the US2016024476 (A1)/US9809801 (B2) and US10828359 (B2)/US2018085449 (A1) patens are in preclinical stage [34,35,36].

The RU2678175 (C1) is related to improving the production of TB/FLU-04L, a vaccine candidate for TB that has been in phase I clinical trials (NCT02501421) since 2013. According to information in the patent description, the novelty of the proposed recombinant strain is its cold-adapted phenotype, which provides additional attenuation and the ability of the strain to actively replicate in the production system. We believe that the patent identified in this review is in the preclinical phase [37,38] (Table 1).

### 3.3. Strength and Limitations of Viral Vector-Based Strategies

As mentioned in this review, viral vectors have several advantages, such as they can carry genes that encode large antigenic fragments; they have stable efficiency in expressing exogenous genes; they induce high levels of cellular and humoral immune responses; immune responses induced by the vector itself can enhance antigen-specific immunological memory; they often do not require adjuvants; they are easy to manipulate and cultivate [24,25,26]. Compared to other vaccine platforms such as mRNA, viral vector vaccines can be more stable, requiring less stringent storage and handling conditions that can range from −25 °C to 8 °C, depending on storage time [26].

However, viral vectors also have limitations. A pertinent characteristic of vector-based vaccines, which is a frequent concern, is recombination, reactogenicity, or reversion to virulence. Therefore, many viral vectors have been genetically modified to make them replication-deficient to increase the safety profile and reduce reactogenicity [25]. In this context, we observed that some patents found in our study presented in their description the assessment of the deficiency of replication of viral vectors and infectivity (US2016024476 (A1)/US9809801 (B2)).

Immunological aspects can also be a source of concern; among them, there are the following: pre-existing immunity, such as neutralizing antibodies against the vector due to previous exposure and immunity may limit the vaccine’s effectiveness; host-induced antiviral immunity can hinder booster vaccination strategies; some viral vectors are not suitable for immunocompromised individuals. An immunization schedule with a vaccine based on viral vectors can induce strong immune responses against the viral vector unrelated to the target antigen. This problem can be solved by using two or more different types of vector vaccines for “primer” and “booster” [24,25,26,111].

In general, side effects ranging from mild, moderate, or strong are commonly reported after vaccination, for example, injection-site pain, redness and swelling, and systemic flu-like symptoms. The recent experience with viral vector vaccines for COVID-19 showed a higher incidence of rare thrombotic and thrombocytopenic cases following administration of viral vector-based anti-SARS-CoV-2 vaccines compared to mRNA-based vaccines [114]. This is a point that will require more understanding for this platform to be used in humans. However, a vaccine candidate for TB based on viral vector, MVA85A, in a randomized, placebo-controlled phase 2b trial, was well tolerated [30].

## 4. Conclusions and Future Directions

The viral vector vaccine strategy has been successful in veterinary medicine as there are some products that have been licensed and in use in the US [115]. On the other hand, until recently, there was only one viral vector-based vaccine approved by the Food and Drug Administration (FDA) for human use, the Ebola vaccine (ERVEBO^®^) [116]. The COVID-19 pandemic has brought several technological platforms to the forefront, and some vaccines based on viral vectors against SARS-CoV-2 obtained emergency use authorization from the WHO and other regulatory agencies [117]. It is important to mention that this platform has been explored not only for prevention but also as a therapeutic delivery system [118,119].

This review provides a general overview of the main strategies explored for the development of vaccines against TB between 2010 and 2023. Focusing on the viral vector vaccine strategy, the review examines the number of patents registered in the period, information on the main types of viral vectors explored, and the most used Mtb antigens and experimental models. A limitation of this study is that, as we used a specific database and search criteria, some patents based on viral vectors for TB and other platforms may not have been covered by this approach; therefore, the number of patents registered between 2010 and 2013 may be higher.

Despite this, there has been a notable increase in interest and application of this technology for the development of TB vaccines in the last 13 years. One of the patents identified in our review is related to TB/FLU-04L, a vaccine candidate that is currently in clinical trials, along with other viral vector-based vaccines for TB, such as MVA85A, ChAdOx1.85A, TB/FLU-01L, and AdHu5Ag85A [120]. Interestingly, none of the vaccines in clinical trials for TB are based on CMV or BAd vectors, although two of the most recently registered patents identified in this review are based on these vectors. For future directions, in the coming years, an increased interest and use of this platform in prophylactic and therapeutic approaches for TB and other diseases is expected. Along with this, expanding knowledge on the safety of this technology is essential to advance its use.

## Figures and Tables

**Figure 1 vaccines-12-00876-f001:**
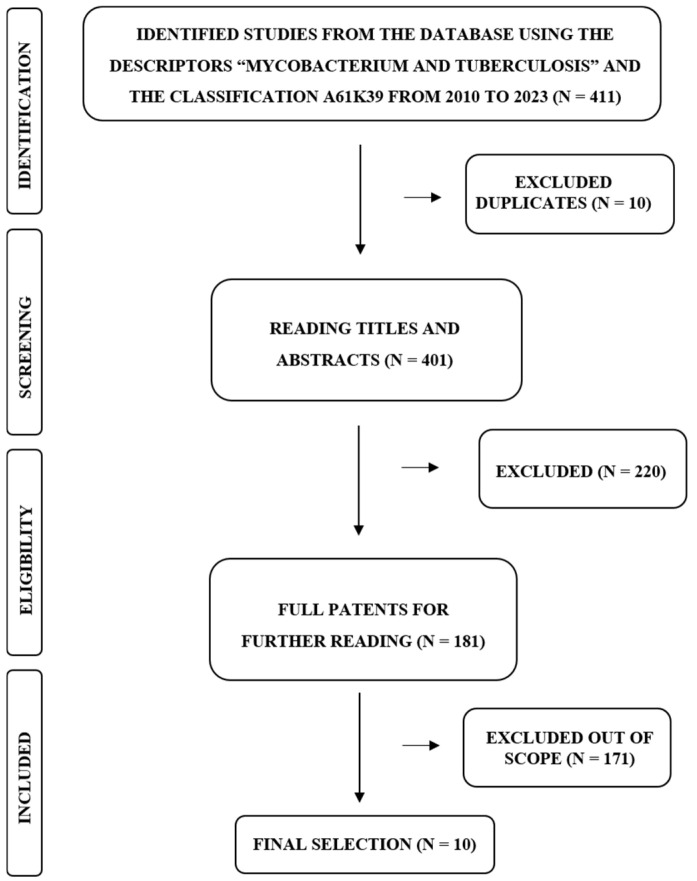
Patent search and screening.

**Figure 2 vaccines-12-00876-f002:**
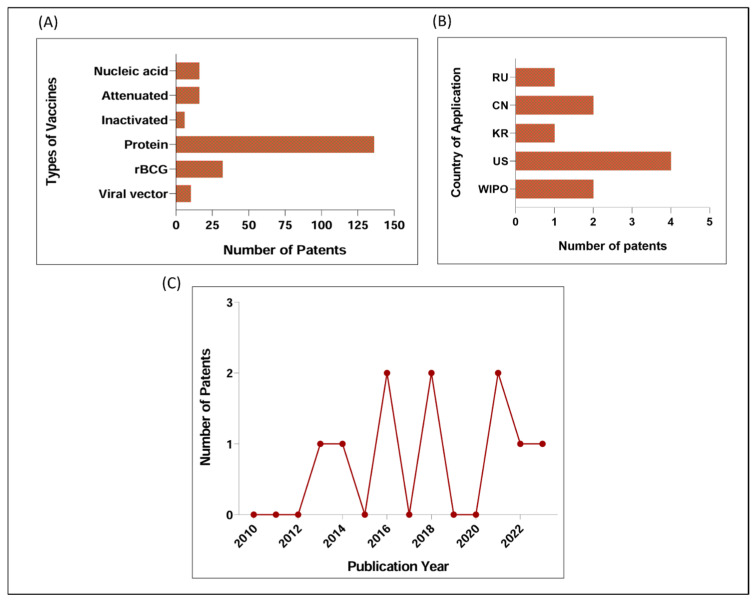
Overview of patents identified through screening: (**A**) Main vaccine strategies identified in patent screening. From 216 patents identified as TB vaccine candidates, around 136 were related to subunit vaccine (protein or fragment-based), 32 to recombinant BCG, 16 to live attenuated, 16 to acid nucleic (DNA or RNA), 10 to viral vector and 6 to inactivated. (**B**) Patents based on viral vector for TB by country of application. Of the 10 selected patents, 4 belong to the United States of America (US), 2 to China (CN), and 2 to the World Intellectual Property Organization (WIPO). Russia (RU) and South Korea (KR) and presented one patent each. (**C**) Number of patents based on viral vector for TB per year of application (2010–2023).

**Table 1 vaccines-12-00876-t001:** Patents based on viral vectors for TB between 2010 and 2023.

Patent Number	Country Year	Viral Vector	Antigen(s)	Methods	Information about Clinical Trials and/or Current Stage of Development
WO2013123579 (A1)	WIPO 2013	Adenovirus-based (chimpanzee adenovirus—AdC)	Antigenic fragments and combinations of Ag85A, Ag85B, TB 10.4, Rv2660c, Rv1773c	BALB/C mice; intranasal route of immunization; evaluation of activated CD8 T cell and IFN-γ-producing cells in the BAL and lung; intratracheal Mtb challenge	Preclinical phase, protective efficacy comparable to or better than BCG [31,32,33]; it was developed by the same group that developed the AdHu5Ag85, which is currently in phase I clinical trials (NCT02337270)
US2016024476 (A1); US9809801 (B2)	USA 2014	Genetically modified arenaviruses (Lymphocytic choriomeningitis virus)	Ag85A, Ag85B, Ag85C, ESAT-6 family (TB10.3, TB12.9 or TB10.4)	C57BL/6 mice; intravenous and subcutaneous routes of immunization; evaluation of CD8+ T cells were measured in peripheral blood, antigen-specific IFN-γ and TNF-α CD8+ and CD4 T cells; replication-deficiency of viral vectors and infectivity assays; Mtb challenge was not performed or not shown	Preclinical phase, protective efficacy comparable to or better than BCG [34]
US10828359 (B2); US2018085449 (A1)	USA 2016	Sendai virus	Antigenic fragments and combinations of Ag85A and Ag85B	BALB/c mice; intranasal and intramuscular routes of immunization; T- cell response and IFN-γ secreted in the lung or spleen; tested alone or as boosting vaccine for BCG; aerosol challenge with H37Rv Mtb strain	Preclinical phase, protective efficacy comparable to or better than BCG [35,36]
CN108018298 (A)	CN 2016	Adenovirus-based (AdHu5)	Lipidated Ag85A	Mice; route of immunization is not clear; antigen-specific IgG antibody titers in mouse serum after immunization; Mtb challenge was not performed or not shown	Preclinial phase; no reference found
KR102135334 (B1); KR20200076335 (A)	KR 2018	Adenovirus-based (AdHu5)	Ag85B, ESAT6, MPT64, Rv2660, and a signal peptide of secretion (tPA)	C57BL/6 mice; evaluated as a booster for BCG vaccine, subcutaneous route of immunization; evaluation of polyfunctional T cells, IFN-γ secretion and humoral response (IgG); challenge with a highly pathogenic Mtb strain (HN878), protective efficacy comparable to or better than BCG	Preclinial phase; no reference found
RU2678175 (C1)	RU 2018	Recombinant influenza virus (influenza A)	ESAT-6 and Ag85A	Related to improving production of TB/FLU-04L (composition already described in another patent); C57BL/6 mice; intranasal route of immunization; evaluation of CD4 and CD8 T cells; Mtb challenge was not performed or not shown	Preclinical phase, protective efficacy comparable to or better than BCG [37,38]; a clinical trial phase I was registered for the TB/FLU-04L (NCT02501421) in 2013
US2021403951 (A1)	USA 2021	Rhesus and Human Cytomegalovirus (recombinant RhCMV or HCMV)	Ag85A-Ag85B-Rv3407, Rv1733-Rv2626c, RpfA-RpfC-RpfD, Ag85B-ESAT-6, and Ag85A-ESAT-6-Rv3407-Rv2626c-RpfA-RpfD	NHP (e.g., Rhesus Macaques); subcutaneous route of immunization; evaluation of CD4 and CD8 T cells in PBMC and BAL; challenge with Erdman Mtb strain; various efficacy criteria evaluated as CT scan analysis, necropsy score, and necropsy Mtb cultures; and 9 extrapulmonary tissues	Preclinical phase, protective efficacy comparable to or better than BCG [39]
CN112899295 (A)	CN 2021	Adenovirus-based (not identified)	Ag85B-ESAT-6 and Rv2031c-Rv2626c	Mice; intranasal route of immunization; evaluation of IgA levels in BAL, antigen-specific antibodies in peripheral blood, spleen lymphocyte proliferation, and tissue memory T cells in BAL; Mtb challenge was not performed or not shown	Preclinical; no reference found
WO2022192163 (A1)	WIPO 2022	Adenovirus-based (AdHu5 and bovine adenovirus—BAd)	Ag85B epitope alone or in fusion with autophagy-inducing peptide C5	C57BL/6 mice; intranasal route of immunization; tested alone or as boosting vaccine for BCG; evaluation of effector and memory T cells after challenge; aerosol challenge with Mtb Erdman strain	Preclinical phase, protective efficacy comparable to or better than BCG [40]
US2023365631 (A1)	USA 2023	Adenovirus-based (AdHu5, chimpanzee adenovirus) and others, such as, poxvirus, RhCMV, HCMV	Ag85B, Ag85A, Rv3407	C57BL/6 and CB6F1 mice; subcutaneous route of immunization; tested as boosting vaccine for BCG or rBCG; evaluation of T cell responses; Mtb challenge was not performed or not shown for adenovirus-based construct	Preclinical phase; no reference found

Abbreviations: WIPO: World Intellectual Property Organization. Countries: USA: United States of America; CN: China; KR: South Korea; RU: Russia.

## Data Availability

The data presented in this study are available on request from the corresponding author.

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
