# Peer review of "Trends in Viral Vector-Based Vaccines for Tuberculosis: A Patent Review (2010–2023)"

_vaccines, 2024, doi:10.3390/vaccines12080876_

Round 1

Reviewer 1 Report

Comments and Suggestions for Authors This review provides a comprehensive overview of the patents related to tuberculosis vaccine candidates, with a specific focus on viral vector-based strategies.   However, it falls short in providing an in-depth discussion regarding the limitations associated with viral vector vaccines. One notable absence is the lack of exploration into potential toxic side effects and immunogenicity upon repeated administration, which are crucial factors to consider in vaccine development.   Addressing these challenges requires a thorough examination of strategies to mitigate adverse effects and enhance vaccine efficacy. Discussing methods to minimize toxicity could provide valuable insights for researchers and policymakers.    Moreover, this review could benefit from presenting a comparative analysis of the advantages and disadvantages of various viral vectors used in tuberculosis vaccine development. A structured comparison in the form of a table would facilitate a clearer understanding of the unique attributes and limitations of each vector platform. This comparative framework could aid researchers in selecting the most suitable vector for their specific vaccine design and highlight areas for further investigation and improvement.   While this review effectively summarizes the landscape of TB vaccine patents, there is a need for deeper analysis and discussion of the challenges associated with viral vector vaccines. By addressing limitations, exploring mitigation strategies, and providing comparative insights, this review could offer valuable guidance for advancing TB vaccine research and development efforts.   In table 1, what does "ESAT-6 e Ag85A" refers to?     Best!

Author Response

Comments 1: This review provides a comprehensive overview of the patents related to tuberculosis vaccine candidates, with a specific focus on viral vector-based strategies.   However, it falls short in providing an in-depth discussion regarding the limitations associated with viral vector vaccines. One notable absence is the lack of exploration into potential toxic side effects and immunogenicity upon repeated administration, which are crucial factors to consider in vaccine development.   Addressing these challenges requires a thorough examination of strategies to mitigate adverse effects and enhance vaccine efficacy. Discussing methods to minimize toxicity could provide valuable insights for researchers and policymakers.   

Response 1: We appreciate the positive appraisal of the manuscript. To address this question, we have included the following topic to the manuscript: “3.3. Strength and limitations of viral vector-based strategies”, lines 501-517.

Comments 2: Moreover, this review could benefit from presenting a comparative analysis of the advantages and disadvantages of various viral vectors used in tuberculosis vaccine development. A structured comparison in the form of a table would facilitate a clearer understanding of the unique attributes and limitations of each vector platform. This comparative framework could aid researchers in selecting the most suitable vector for their specific vaccine design and highlight areas for further investigation and improvement.   While this review effectively summarizes the landscape of TB vaccine patents, there is a need for deeper analysis and discussion of the challenges associated with viral vector vaccines. By addressing limitations, exploring mitigation strategies, and providing comparative insights, this review could offer valuable guidance for advancing TB vaccine research and development efforts.  

Response 2: We thank the reviewer for this suggestion. In addition to adding the topic “3.3. Strength and limitations of viral vector-based strategies”, lines 485-517; we have added throughout the manuscript a comparison of the vectors, focusing mainly on the advantages considering RNA and DNA-based vectors, and also on the high pre-existing immunity against some vectors more than others; lines, 444-457 and 469-477.

Comment 3: In table 1, what does "ESAT-6 e Ag85A" refers to?     Best!

Response 3: Thanks to the reviewer for pointing this out. The correct one is "ESAT-6 and Ag85A" and we have changed in the table.

Reviewer 2 Report

Comments and Suggestions for Authors

In their review, Camelo dos Santos and colleagues focus on 10 viral-based vaccines, currently in development, to fight TB. They also describe the antigens and the experimental models used in the corresponding studies. As such vaccines show some promise, this review may be of interest. However, several references need to be corrected. In addition, the authors often use indirect references (see below), which should be avoided.

Line 40: Is reference 1 correct? It does not seem to be appropriate to introduce TB as a global burden.

Lines 44-46: As the Global TB Report 2023 has been published, the reference 3 needs to be updated.

Lines 64-66: Reference 9 cites the Global TB Report 2020. Again, there are updates that are more recent updates to the Global TB Report.

As a general comment, please avoid citing indirect sources in your review (e.g. the reference 9 instead of the Global TB Report 2020). Therefore, check for all indirect references and correct them in the manuscript.

Lines 71-73: Are there no better references to measure the impact of BCG vaccination? Furthermore, the main topic of the reference 12 has nothing to do with BCG vaccination. It is again an indirect reference.

Lines 82-84: Please check the accuracy of the whole list. For example, there is no direct relationship between H4:IC31 (reference 26) and recombinant BCG vaccines. Reference 38 is a general review with no direct link with to viral vector vaccines.

Figure 1: Reeding? Keyworks?

Figure 2A: 216 patents were identified as TB vaccine candidates, but Figure 1 shows that only 181 patents were eligible.

Figure 2B: KR vs KO. There is a strange numbering in the legend (and throughout the manuscript), e.g. 04 instead of 4.

Table 1: What do A1, B2 and so on mean? Why are some antigens placed in brackets? EAST-6 e Ag85A?

Lines 200-202: Reference 46 is very specific and does not describe adenoviruses in general.

Lines 317-319: Does the reference 94 really describe the cord factor?

Line 322: SIgA is not defined.

Comments on the Quality of English Language

Minor editing of English language required

Author Response

Comments 1: In their review, Camelo dos Santos and colleagues focus on 10 viral-based vaccines, currently in development, to fight TB. They also describe the antigens and the experimental models used in the corresponding studies. As such vaccines show some promise, this review may be of interest. However, several references need to be corrected. In addition, the authors often use indirect references (see below), which should be avoided.

Response 1: We completely agree and thank the reviewer for highlighting this. We have reviewed all references to address this.

Comments 2: Line 40: Is reference 1 correct? It does not seem to be appropriate to introduce TB as a global burden.

Response 2: We agree with the reviewer and have changed the reference.

Comments 3: Lines 44-46: As the Global TB Report 2023 has been published, the reference 3 needs to be updated.

Response 3: We agree with the reviewer and have updated the reference.

Comments 4: Lines 64-66: Reference 9 cites the Global TB Report 2020. Again, there are updates that are more recent updates to the Global TB Report.

As a general comment, please avoid citing indirect sources in your review (e.g. the reference 9 instead of the Global TB Report 2020). Therefore, check for all indirect references and correct them in the manuscript.

Response 4: We agree with the reviewer and have revised all the references. Please note that as we have reviewed all references, the reference number is now 3, lines 66-67.

Comments 5: Lines 71-73: Are there no better references to measure the impact of BCG vaccination? Furthermore, the main topic of the reference 12 has nothing to do with BCG vaccination. It is again an indirect reference.

Response: We agree with the reviewer and have changed the reference. Please note that as we have reviewed all references, they may have changed the number, lines 74-79.

Comments 6: Lines 82-84: Please check the accuracy of the whole list. For example, there is no direct relationship between H4:IC31 (reference 26) and recombinant BCG vaccines. Reference 38 is a general review with no direct link with to viral vector vaccines.

Response: We agree with the reviewer and have revised all the references. Please note that as we have reviewed all references, they may have changed the number, lines 83-87.

Comments 7: Figure 1: Reeding? Keyworks?

Response: Thanks to the reviewer for pointing this out. We have corrected the words in the Figure 1: “reading” and “descriptors”.

Comments 8: Figure 2A: 216 patents were identified as TB vaccine candidates, but Figure 1 shows that only 181 patents were eligible.

Response: 181 patents were selected for full reading (only protein subunit vaccines were included at this stage – eligibility), as explained in the methods section, lines 115-122. Figure 2A was constructed based on our search in the “screening and eligibility stages”, thus the number of patents is different because were evaluated in different stages.

Comments 9: Figure 2B: KR vs KO. There is a strange numbering in the legend (and throughout the manuscript), e.g. 04 instead of 4.

Response: Thanks to the reviewer for pointing this out. We have made the corrections as suggested, line 147.

Comments 10: Table 1: What do A1, B2 and so on mean? Why are some antigens placed in brackets? EAST-6 e Ag85A?

Response: We appreciate the careful analysis of the manuscript. The “A1” and “B2” are part of the patent identification code as described in the database. We use the square brackets to specify, if necessary, for example TB10.3, TB12.9 or TB10.4 are part of the ESAT-6 family. We have included the term “family” to make it clearer. The correct one is “ESAT-6 and Ag85A” and we have changed it in the table 1.

Comments 11: Lines 200-202: Reference 46 is very specific and does not describe adenoviruses in general.

Response: We agree with the reviewer and have changed the reference. Please note that as we have reviewed all references, the reference number is now 42.

Comments 12: Lines 317-319: Does the reference 94 really describe the cord factor?

Response: We appreciate the careful analysis of the manuscript and have changed the reference. Please note that as we have reviewed all references, they may have changed the number, line 347 (references 81, 82).

Comments 13: Line 322: SIgA is not defined.

Response: We have added the description: secretory immunoglobulin A (SIgA), line 350.

Reviewer 3 Report

Comments and Suggestions for Authors

The authors have reviewed the patents filed for viral vector-based TB vaccines between the years 2010 and 2023. Compared to the all patents filed for TB vaccines during this time (more than 400), there were only 10 patents for viral vector-based TB vaccines.

The authors discuss in detail the antigens and the viruses used in these patents as well as the animal models used for evaluation. This detailed information is only of interest to the readers, if these vaccines provide better protection than the currently used vaccines. This information is missing.

Since filing a patent is different from advancing to a vaccine, it would be of interest to readers to get more information about the further development, performance in clinical trials, efficacy and actual use as vaccines of these patents. I think the authors should expand on this.

The authors state that they expect an increased use of viral vector-based TB vaccines. They should state what is the basis for this expectation. There have been a number of publications mentioning that some viral vector-based TB vaccines are already in clinical trials, e.g. reviewed by Ottenhoff and Kaufmann (2012), Roman et al. (2023). Some of these vaccines were more than 10 years ago in phase I and phase IIb clinical trials. It would be of interest to the reader to see how these vaccines performed and for what purpose they were used (pre-infection, post-exposure, age groups, generalized/pulmonary TB, booster).  I am not sure, if there is any viral vector-based TB vaccine currently in use. If there is none in use, the authors should include information about problems encountered – which might prevent the further development of the patents as TB vaccines. In the discussion, an explanation is missing why the majority of patents favor other platforms.

Finally, the authors might want to consider and include a comparative aspect: viral vector-based TB vaccines have been developed and tested for bovine TB. In this context, they were used as booster vaccines after BCG immunization and they were tested in more appropriate models than mice (e.g. Vordermeier et al. 2014).

Comments on the Quality of English Language

There are a few spelling errors, e.g. L23 to fill is not to file. 

Sometimes, I am not sure if the correct word is used, e.g. L78 enduring = long-lasting? prolonged?

This should be checked by a native speaker (which I am not!!).

Author Response

Comment 1: The authors have reviewed the patents filed for viral vector-based TB vaccines between the years 2010 and 2023. Compared to the all patents filed for TB vaccines during this time (more than 400), there were only 10 patents for viral vector-based TB vaccines.

Response 1: We appreciate the positive appraisal of the manuscript.

Comment 2: The authors discuss in detail the antigens and the viruses used in these patents as well as the animal models used for evaluation. This detailed information is only of interest to the readers, if these vaccines provide better protection than the currently used vaccines. This information is missing.

Response 2: Thanks to the reviewer for bringing this to light. The manuscript is mainly based on the information we were able to identify in the patent description. Unfortunately, some information may not be available in the description. For example, as shown in Table 1, we included information about whether Mtb challenge was performed (protection), and for five patents “Mtb challenge was not performed or was not shown” in the patent description. Although we understand that this information is important and have included it in the table when available in the patent or reference (as we have expanded Table 1).

Comment 3: Since filing a patent is different from advancing to a vaccine, it would be of interest to readers to get more information about the further development, performance in clinical trials, efficacy and actual use as vaccines of these patents. I think the authors should expand on this.

Response 3: We thank the reviewer for this suggestion. We have included another column in Table 1, called “Information about clinical trials and/or current stage of development”; and included a topic in the manuscript, “3.2.4. Current stage of patent development: information on preclinical and clinical trials”, lines 433-480.

Comment 4: The authors state that they expect an increased use of viral vector-based TB vaccines. They should state what is the basis for this expectation. There have been a number of publications mentioning that some viral vector-based TB vaccines are already in clinical trials, e.g. reviewed by Ottenhoff and Kaufmann (2012), Roman et al. (2023). Some of these vaccines were more than 10 years ago in phase I and phase IIb clinical trials. It would be of interest to the reader to see how these vaccines performed and for what purpose they were used (pre-infection, post-exposure, age groups, generalized/pulmonary TB, booster).  I am not sure, if there is any viral vector-based TB vaccine currently in use. If there is none in use, the authors should include information about problems encountered – which might prevent the further development of the patents as TB vaccines.

Response 4: We appreciate the careful analysis of the manuscript and have included a paragraph to address this, lines 182-196. Regarding the question of whether there is a viral vector-based vaccine for TB in use, the only vaccine available for use in humans against TB is the BCG.

Comment 5: In the discussion, an explanation is missing why the majority of patents favor other platforms.

Response 5: We thank to the reviewer for pointing this out. We have included a comment on this, lines 160-166.

Comment 6: Finally, the authors might want to consider and include a comparative aspect: viral vector-based TB vaccines have been developed and tested for bovine TB. In this context, they were used as booster vaccines after BCG immunization and they were tested in more appropriate models than mice (e.g. Vordermeier et al. 2014).

Response 6: We thank the reviewer for this suggestion. Although we did not do an in-depth comparative analysis because it would take more time and perhaps, we would be going a little beyond the objective of the manuscript, we understand that a comment on this would improve the manuscript and include, lines 425-432.

English checking

There are a few spelling errors, e.g. L23 to fill is not to file.

Response: Thanks to the reviewer for pointing this out. We have corrected, line 25.

Sometimes, I am not sure if the correct word is used, e.g. L78 enduring = long-lasting? prolonged? This should be checked by a native speaker (which I am not!!).

Response: We agree with the reviewer and have changed to “longer-lasting”, line 81.

Reviewer 4 Report

Comments and Suggestions for Authors

In the current manuscript the authors report on the results of their patent review focused on viral vector-based vaccines against tuberculosis. The design of the authors patent review is sound and well-described in the manuscript. Likewise, the authors' presentation of the results of their review is understandable and detailed. The authors also did a thorough job discussing the various TB vaccine platforms and vaccine antigens revealed in their patent review. My main critique is that this study is primarily descriptive; there is minimal discussion on the future of TB vaccines and the potential for the development of an effective vaccine.

1. The patents the authors find in their review appear to be overwhelming at the experimental phase. It is unclear which of these have advanced to clinical trials, or have failed to progress. It would be helpful to note this in Table I (the authors mention in line 148 that 5 vaccines had progressed to clinical trials).

2. The clinical potential for each vaccine in Table I might also be made apparent by the number of publications and the year of the most recent publication.

3. How the TB vaccine field views the potential of viral vector-based vaccines might be revealed by plotting other patents in panel C of Figure 2. For example, has the number of recent viral vector-based vaccine patents been eclipsed by other modalities or has it stayed consistent/competitive?

Comments on the Quality of English Language

There appears to be a spelling error in Figure 1. Presumably "Reeding Titles and Abstracts" listed in the Screening portion of the flowchart should read "Reading Titles and Abstracts".

Author Response

Comment: In the current manuscript the authors report on the results of their patent review focused on viral vector-based vaccines against tuberculosis. The design of the authors patent review is sound and well-described in the manuscript. Likewise, the authors' presentation of the results of their review is understandable and detailed. The authors also did a thorough job discussing the various TB vaccine platforms and vaccine antigens revealed in their patent review. My main critique is that this study is primarily descriptive; there is minimal discussion on the future of TB vaccines and the potential for the development of an effective vaccine.

Response: We appreciate the positive appraisal of the manuscript. We agree that it is a mainly descriptive work, but we believe that it provides an overview of the main strategies of interest in the last 13 years (as there are patents filed), as well as the detailed information on patents based on viral vectors can provide valuable information for researchers and policy makers in the context of the development of vaccines against TB. For example, it is possible to note that active or well-known immunodominant Mtb antigens have been extensively applied in the development of TB vaccines, and that in recent years different antigens are being explored, Table 1 and lines 331-341.

Comment 1. The patents the authors find in their review appear to be overwhelming at the experimental phase. It is unclear which of these have advanced to clinical trials, or have failed to progress. It would be helpful to note this in Table I (the authors mention in line 148 that 5 vaccines had progressed to clinical trials).

Response 1: We thank the reviewer for this suggestion. We have included another column in Table 1, called “Information about clinical trials and/or current stage of development”; and included a topic in the manuscript, “3.2.4. Current stage of patent development: information on preclinical and clinical trials”, lines 433-480.

Comment 2. The clinical potential for each vaccine in Table I might also be made apparent by the number of publications and the year of the most recent publication.

Response 2: We agree with the reviewer and have included another column in Table 1, called “Information on clinical trials and/or current stage of development”. This column also has information about references/publications that we could identify related to the patents.  As mentioned before, we also have included a topic in the manuscript, “3.2.4. Current stage of patent development: information on preclinical and clinical trials”, lines 433-480.

Comment 3. How the TB vaccine field views the potential of viral vector-based vaccines might be revealed by plotting other patents in panel C of Figure 2. For example, has the number of recent viral vector-based vaccine patents been eclipsed by other modalities or has it stayed consistent/competitive?

Response 3: We thank the reviewer for this suggestion. Although we agree that this would be a different way of looking at the data, it would require an extensive analysis based on the number of patents in the “screening and eligibility phase” of this study's methodology (approximately 216 patents would have to be evaluated per year). Furthermore, as we found few patents based on viral vectors, perhaps this analysis would not add more information for comparison between the different strategies.

English checking

There appears to be a spelling error in Figure 1. Presumably "Reeding Titles and Abstracts" listed in the Screening portion of the flowchart should read "Reading Titles and Abstracts".

Response: Thanks to the reviewer for pointing this out. We have corrected.

Round 2

Reviewer 2 Report

Comments and Suggestions for Authors

The authors addressed all my concerns.

Line 428: 108refs?

Comments on the Quality of English Language

Minor editing of English language required

Author Response

Comment: The authors addressed all my concerns. Line 428: 108refs?

Response: We thank the reviewer for pointing this out and careful analysis of the manuscript. We have correct.

Reviewer 3 Report

Comments and Suggestions for Authors

The authors have addressed all my concerns and comments.

Could you please check in Table 1 on page 26: are the first 3 lines in the correct column?

Author Response

Comment: The authors have addressed all my concerns and comments. Could you please check in Table 1 on page 26: are the first 3 lines in the correct column?

Response: We thank the reviewer for the careful analysis of the manuscript, we checked Table 1 and it is OK. We couldn't notice errors in the rows and columns.